# Enhancing user perception and drone flight control through AR-based object projection

**Runa Sawayama**[ID][☉]*, **Hajime Nobuhara**[☉]

University of Tsukuba, Tsukuba, Ibaraki, Japan

☉ These authors contributed equally to this work.
* sawayama@cmu.iit.tsukuba.ac.jp

## Abstract

This study explores the question: "Can changing the appearance of drones through AR influence user operational behavior and impressions of drones?" To the best of our knowledge, this is the first study to empirically examine the influence of AR-based appearance modification of drones on user perception and operational behavior. To investigate this, a method is proposed to modify the appearance of drones by projecting objects onto them using Augmented Reality (AR), and its effects are evaluated through subjective experiments involving a relatively large number of participants. Data for drone operation and survey responses are collected from participants and analyzed using factor analysis. The factor scores are then compared between the conditions with and without AR-based appearance modification. The results do not indicate significant differences in factors related to the number of operations, suggesting consistency in this aspect. However, significant differences are observed in factors related to appearance and social evaluation (age, animal likeness, gender), as well as in physical characteristics (e.g., size, weight) of drones. In addition, open-ended survey responses reveal notable differences in appearance assessment. These findings demonstrate the potential of this method to improve the social acceptance of drone technology, particularly in the context of this research.

## Introduction

Human perception is heavily influenced by vision. Studies have shown that manipulating an object's appearance using Augmented Reality (AR) can alter other sensory experiences, such as smell and touch [1–4]. This demonstrates the critical role of appearance in shaping user perception.

With the increasing adoption of drones in various fields, research has found that their appearance can cause fear among users [5]. Motivated by the lack of research on real drones, this study experimentally investigates how AR-modified drone appearances influence both user operational behavior and impressions, aiming to clarify the effects of AR-based appearance modification in real-time drone interaction.

provided the original author and source are credited.

**Data availability statement:** Data cannot be shared publicly because the dataset contains potentially identifiable personal information from human participants. The data are not publicly available due to participant confidentiality. The data are available upon reasonable request for researchers who meet the criteria for access to confidential data. Requests for data access should be directed to the Research Ethics Committee, Faculty of Engineering, Information and Systems, University of Tsukuba, which imposed the restrictions on the data. Contact information: submit3@un.tsukuba.ac.jp. The authors are not able to share the data directly due to ethical and legal restrictions.

**Funding:** The author(s) received no specific funding for this work.

**Competing interests:** The authors have declared that no competing interests exist.

For example, Xu et al. (2025) [6] proposed SafeSpect, a safety-first AR heads-up display for drone inspections, demonstrating that AR interfaces can reduce cognitive load and enhance situational awareness for drone pilots. Drones are typically designed with a symmetric shape to enhance flight control. However, this symmetry can make it difficult for users to determine their direction of movement. As a result, there is a recognized need for designs that clearly communicate the direction and speed of drone movement to users [5]. Although extensive research has explored AR/VR applications for supporting human–machine interaction and task performance in industrial systems [7,8], these studies primarily focus on static environments, virtual objects, or head-mounted interfaces, rather than dynamically modifying the appearance of real, fast-moving robotic agents such as drones.

Despite this, dynamically altering the physical appearance of drones in real time is challenging, especially given their high-speed movement. Consequently, little research has been done on how changing drone appearances through AR could impact user impressions or operational behavior. To address this gap, this study poses the following research question: "**Can changing the appearance of drones through AR influence user operational behavior and impressions of drones?**"

As this work represents an initial investigation into this topic, the study is positioned as an exploratory study aimed at identifying potential effects and informing future research.

Therefore, the objective of this research is: "**To clarify the impact of altering drone appearances through AR on user operational behavior and impressions by conducting experiments with a large number of participants.**"

An overview of the proposed AR-based appearance modification framework and the experimental design is shown in Fig 1.

By constructing this AR environment and conducting experiments, we analyze the controller operation data collected and the responses to the questionnaire using factor analysis. We compare factor scores between conditions where the drone's appearance is altered and unchanged. The results show no significant differences in operational inputs, but reveal significant differences in factors related to social perception and physical characteristics. These findings suggest that the appearance of drones can be modified to enhance social and physical impressions without negatively impacting user operation.

The key contributions of this research are as follows.

1. Development of a projection framework capable of tracking small high-speed drones.

2. Quantitative analysis of operational inputs using marker detection on controllers, based on flight operations from 21 participants.

3. Empirical evidence that changing the appearance of a drone does not affect operational behavior but can improve the user impressions of drones.

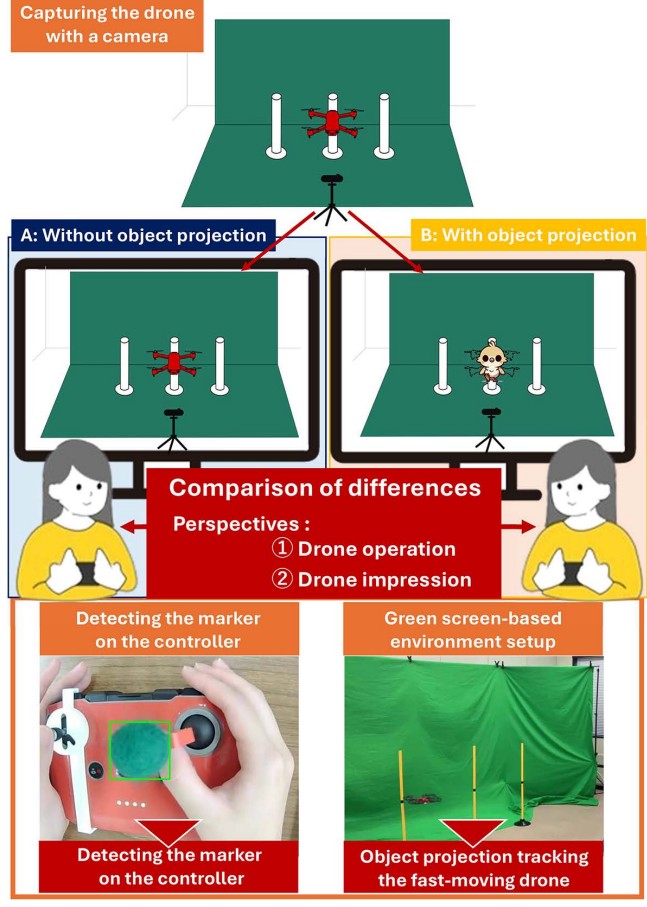

**Fig 1. Research summary.** Overview of a system where the projected object changes in response to drone controller inputs and an experiment investigating the psychological effects on users.

The structure of this paper is as follows: Section Related Work reviews related work. Section Proposed Methodology describes the proposed investigation methodology. Section Results presents the results, and Section Conclusion concludes the study.

## Related work

This section reviews prior studies on AR-based appearance modification and drone-related interfaces, and clarifies the positioning of the present study.

### Research on appearance changes using AR

AR has been widely used to modify the perceived appearance of objects and to investigate cross-modal perception. For example, prior work has altered the appearance of food or tableware to influence taste perception [1–3,9,10].

Similarly, studies have shown that visual modifications can influence tactile perception, such as perceived softness [4] and weight [11]. Appearance manipulation has also been linked to changes in satiety perception [9] and self-related perceptions, including body satisfaction and self-concept [12–14].

These findings demonstrate that AR-based appearance manipulation can systematically influence human perception. However, its influence on operational behavior in dynamic human–machine interaction contexts remains underexplored.

### Research on drone control, appearance, and AR interfaces

Drone control research has explored alternatives to traditional controllers, including gaze-based control [15] and gesture or body-based interaction [16].

AR applications in drone systems typically focus on navigation support, such as 3D reconstruction [17], telemetry visualization [18,19], and first-person displays [20,21]. Xu et al. [6] proposed an AR interface that overlays safety information to reduce cognitive load during inspections.

Regarding drone appearance, prior studies have examined custom drone designs or aesthetic evaluations [5,22,23]. However, these works mainly rely on subjective evaluations and do not investigate how appearance influences operational behavior.

Technology acceptance studies in drones often use theoretical acceptance models [24–26], with limited empirical data from real flight operations.

### Positioning of this study

Object appearance is known to affect approachability and usability [5,27]. While prior research has examined virtual drones or static appearance design, studies that dynamically modify the appearance of real, fast-moving drones using AR and quantitatively evaluate both operational behavior and user impressions are lacking.

This gap likely reflects the technical difficulty of real-time tracking and projection onto moving drones. The present study addresses this challenge by developing a projection framework capable of modifying drone appearance during flight and empirically evaluating its psychological and behavioral effects. This positions the work as a novel contribution at the intersection of AR and drone interaction research.

However, prior research has focused primarily on operational analysis and appearance evaluation using virtual drones. Studies that employ AR to dynamically alter the appearance of actual, high-speed drones and quantitatively measure its effects on both operational behavior and user impressions are lacking. This gap likely arises from the technical challenges involved in real-time AR-based appearance modification and tracking, which our study specifically addresses.

To address these challenges, this study constructs an environment to improve drone detection accuracy and employs AR technology to change drone appearances. Experiments are conducted to examine the impact of these changes on operational behavior. This approach represents a novel development in both drone and AR research fields.

## Proposed methodology

This section describes the method for quantitatively measuring user controller input during a drone operation experience enhanced by AR-projected objects, as well as the experimental design that leverages this method.

### Overview of the AR system

Fig 2 illustrates the overview of the AR system developed. The flight of the drone is captured by a camera, and the drone is detected from the video footage. To facilitate detection of the drone, which is small and moves quickly, red stickers are applied to its entire surface. In addition, a green screen is used as background to improve the detection accuracy.

In this system, objects are projected onto the detected position of the drone within the captured video. These projected objects change their behavior according to the controller's input, allowing the operator's intentions to be reflected in the drone's movements. To quantify controller input, markers are attached to the controller sticks and the input motions are recorded from above. Marker positions are used to convert the input from the controller into measurable values.

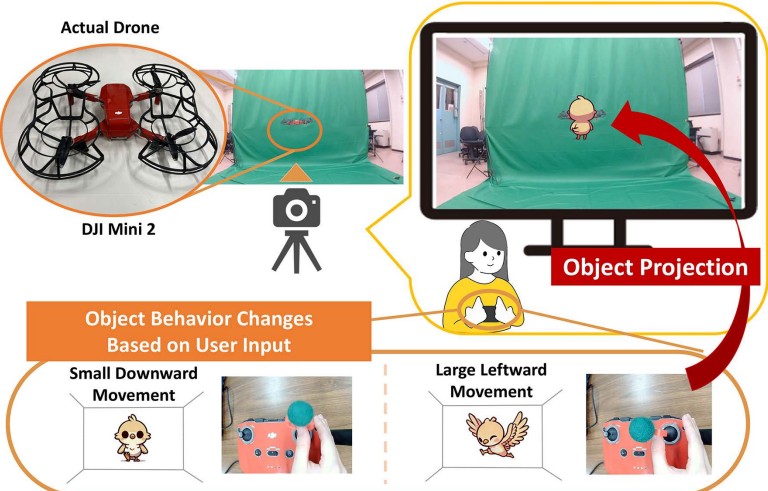

**Fig 2. Overview of the proposed AR system.** This AR system captures the drone's flight using a camera, detects the drone from the video footage, and projects objects onto it.

## Creation of projected objects

To visually present the drone's movement to the operator, objects projected in response to the controller input are described here.

Markers are attached to the right stick of the controller, which is used to move the drone forward, backward, left, and right. The controller's motion is recorded from above by a camera. The magnitude of movement $L$ ($\in [0, 1]$) is defined as the ratio of the distance from the central position to the marker's current position to the maximum distance when the controller is fully moved. The direction $\theta$ is classified as follows, with the right direction defined as $\theta = 0°$ :

- Right: $-45° < \theta < 45°$

- Up: $45° < \theta < 135°$

- Left: $135° < \theta < 225°$

- Down: $-135° < \theta < -45°$

Using these metrics, the controller input is quantified by direction and magnitude.

Fig 3 illustrates how the controller stick input direction and magnitude are detected based on the position of the marker attached to the controller.

As shown in Fig 4, the projected object's behavior is modified based on the direction and magnitude of the controller stick's movement detected by the marker position. A bird-shaped character, as illustrated in Fig 4, was selected as the projected object for the following reasons:

Animal-inspired designs have been suggested to promote social interaction with humans, increasing familiarity and likability [27]. In addition, cartoon-like appearances have been reported to emphasize playfulness and approachability, receiving higher evaluations compared to mechanical appearances [28]. By projecting a bird-shaped character, this study aims to enhance user familiarity and reduce psychological resistance, such as fear.

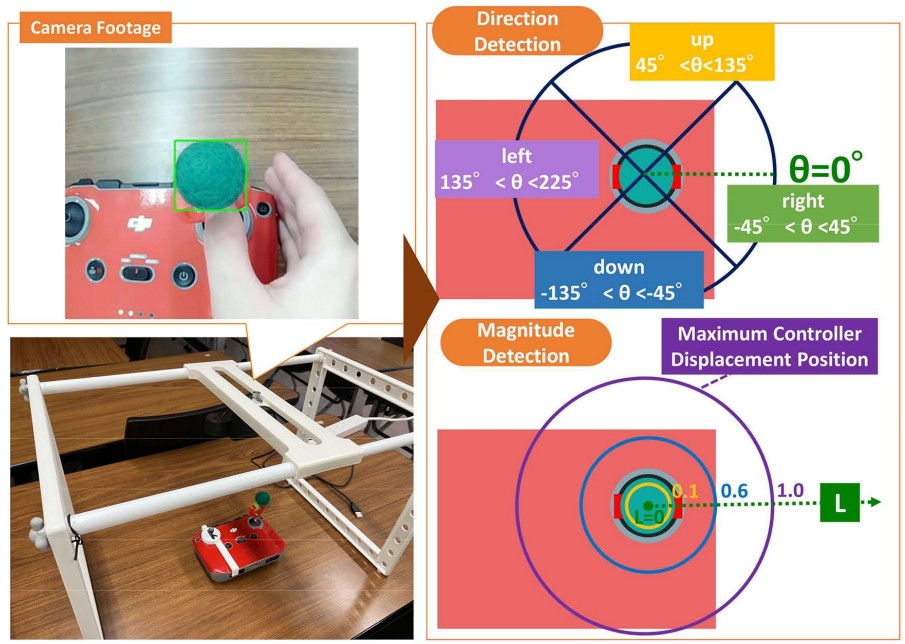

**Fig 3. Detection of controller direction and input magnitude.** Detecting the input direction and magnitude based on the position of the marker attached to the controller stick.

## Experimental environment

The experimental environment is shown in Fig 5. The participants operated the drone within an area captured by the camera, located 2.5 meters away. The operable area of the drone is $1.2m$ (height) × $5.0m$ (width) × $2.0m$ (depth).

The DJI Mini2, as shown in Fig 6, was used in this experiment. This drone is compact and lightweight, with a diagonal dimension of 233 mm and a weight of 199 g, making it suitable for indoor operations. To facilitate the detection of the small, fast-moving drone, red stickers were applied to its surface, enabling the AR system to identify it as a red object. The green screen environment further improved the detection accuracy and stability. In designing the experiment, we faced a trade-off between flight safety and AR detection precision. While a smaller drone was preferable for ensuring participant safety, our color-based detection method meant that a larger drone could act as a larger marker, enhancing AR alignment and projection accuracy. Considering this trade-off, the DJI Mini2 was selected as an optimal compromise between safety and tracking stability. Furthermore, environmental settings, including green screen placement, and the drone's color configuration were carefully adjusted to maximize AR-based appearance modification accuracy.

Fig 7 shows an example of the AR system screen during drone operation. The participants operated the drone while observing the projected object displayed in the video.

## Overview of the experimental method

This section describes the experimental conditions for 21 participants ($n = 21$). The participants comprised 12 males and 9 females, aged between 18 and 50 years (mean $= 24.9$, variance $= 52.5$). Seven participants had prior experience using drones, whereas 14 had no such experience. Although the number of participants was limited to 21, this sample size was considered sufficient for an exploratory study. Similar to Umar (2016) [29], who conducted qualitative research on safety

| Direction | Magnitude $L$ | Projected Object | Direction | Magnitude $L$ | Projected Object |
|---|---|---|---|---|---|
| — | ~0.1 | | up | 0.6~ | |
| up | 0.1 ~ 0.6 | | down | 0.6~ | |
| down | 0.1 ~ 0.6 | | left | 0.6~ | |
| left | 0.1 ~ 0.6 | | right | 0.6~ | |
| right | 0.1 ~ 0.6 | | | | |

**Fig 4. Correspondence between controller input and projected object.** The behavior of the projected object is modified as shown in this figure based on the direction and magnitude of the controller stick input.

climate in the Omani construction industry with six participants, small samples can provide rich and valuable insights into the research questions. The procedure of the proposed experimental method is shown in Fig 8.

The experimental procedure is as follows:

(1) Participants complete a questionnaire about their attributes and impressions of drones.

(2) Participants practice operating a drone for five minutes using a monitor.

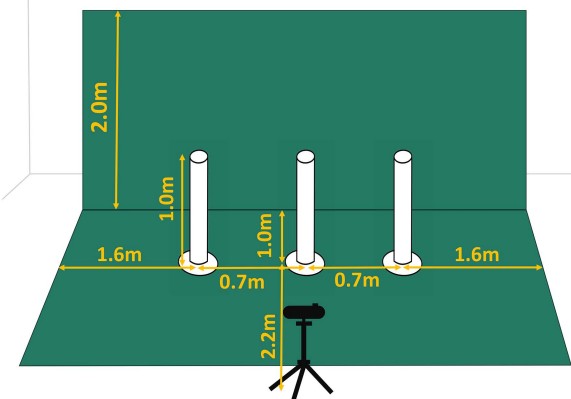

**Fig 5. Experimental environment.** Overview of the experimental environment. A green screen is used as the background, and three poles are set up. The drone's operating area is defined as 1.2m (height) × 5.0m (width) × 2.0m (depth).

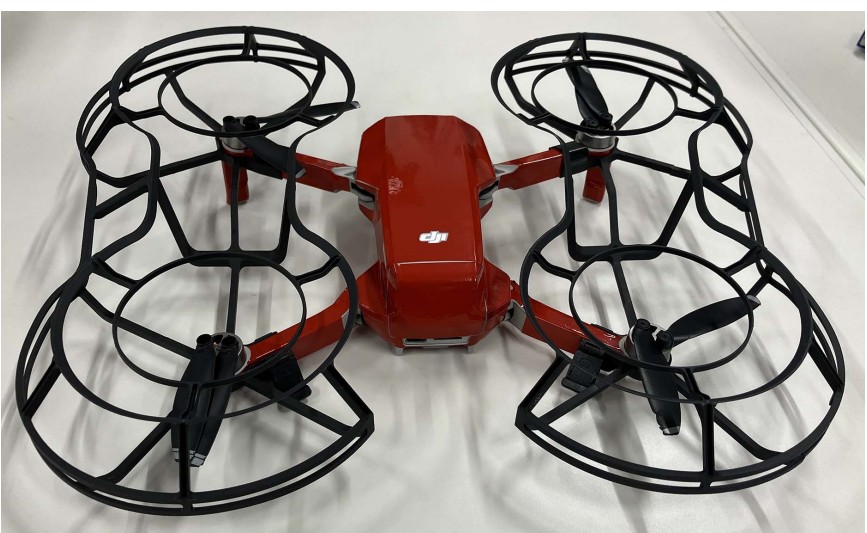

**Fig 6. Modified commercial drone (DJI Mini2) for improved detection.** Red stickers are applied to the surface of the compact and lightweight DJI Mini2.

(3) Using the AR system implemented in Section Overview of the AR system, participants complete three rounds of drone operation tasks and answer a questionnaire to evaluate their impressions under the following conditions:

1. **Condition A:** Operate the drone without any projections.

2. **Condition B:** Operate the drone with projected objects.

To prevent the order of conditions from affecting the results, the sequence of conditions was counterbalanced among participants. Half of the participants ($n = 10$) performed Condition A first, followed by Condition B, while the other half ($n = 11$) performed Condition B first, followed by Condition A.

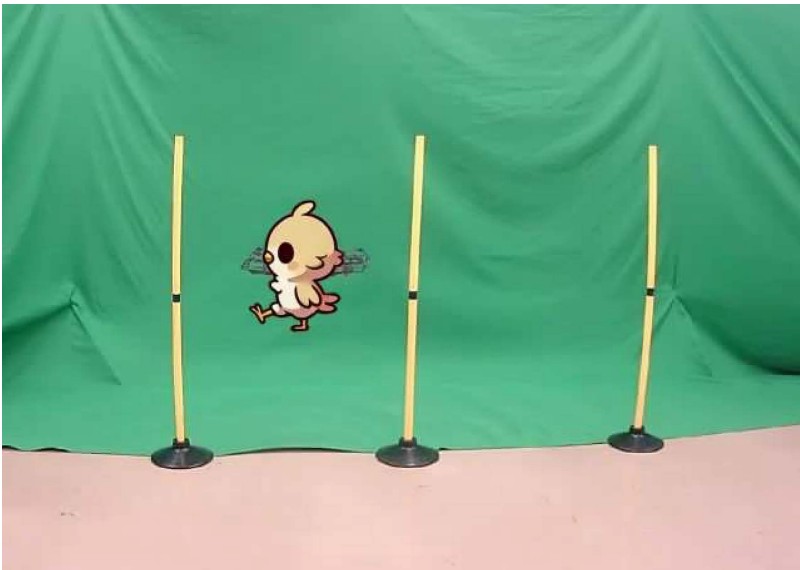

**Fig 7. Example AR system screen.** Example AR system screen during drone operation. Participants operate the drone while observing the projected object displayed on the video.

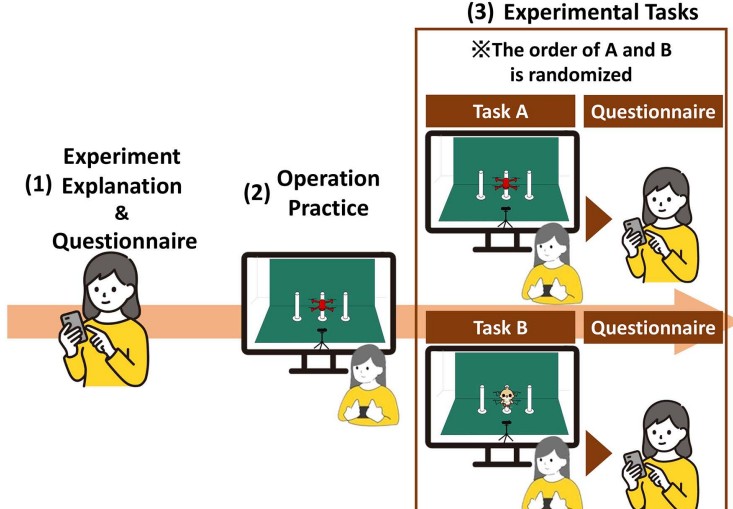

**Fig 8. Experimental procedure.** This figure illustrates the experimental procedure. After completing a preliminary questionnaire, participants practiced operating a drone for five minutes. They then performed three drone operation tasks using the AR system, followed by a final questionnaire.

The results of the questionnaire and the video recordings of drone operations were analyzed to investigate how AR-projected objects influence the impressions of participants of drones and their operational behavior. This experiment was carried out with the approval of the Ethics Review Committee (Ethics Review Number: 2024R867). Participants were recruited between September 15 and September 30, 2024, and all participants provided written informed consent prior to participation.

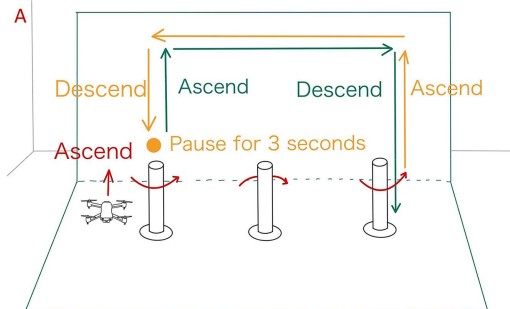

## Design of the experimental tasks

This section describes the drone operation tasks performed by the participants in Section Overview of the Experimental Method.

Figs 9 and 10 illustrate the drone operation tasks performed under conditions A and B, respectively. These tasks were designed with reference to related studies on drone operation tasks [15] and include the following:

1. **Left Stick of the Controller:** Used to control vertical movements (ascending and descending).

2. **Right Stick of the Controller:** Used to control horizontal movements (forward, backward, left, and right), including the following tasks:

   • Moving forward while avoiding poles.

   • Performing wide lateral movements.

   • Stable hovering.

Since this experiment focuses on the planar horizontal movement of the drone, the rotational movements controlled by the left stick (yaw rotation) were disabled. The left stick was restricted to vertical movement only. Each task shown in Figs 9 and 10 was repeated three times under each experimental condition.

Furthermore, the operation tasks for Conditions A and B were designed symmetrically to minimize the impact of task differences on the analysis results.

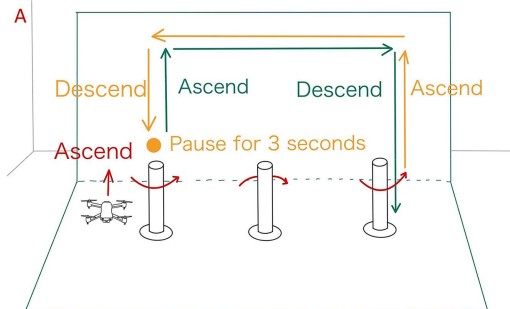

**Fig 9. Drone operation task for condition A.** Task performed under the condition without projecting objects onto the drone.

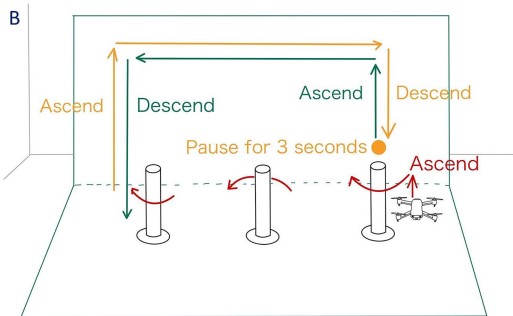

**Fig 10. Drone operation task for condition B.** Task performed under the condition with projecting objects onto the drone.

## Design of the questionnaire

This section describes the questionnaire conducted in Section Overview of the Experimental Method. The items used in the questionnaire are shown in Fig 11. Participants responded to these items using a 7-point Likert scale.

The questionnaire items were categorized into three main groups:

1. **Preliminary Investigation of Drone Impressions:** These items were identified through a preliminary investigation aimed at elucidating differences in drone perceptions before and after use, revealing their influence on users' willingness to adopt the technology. This study seeks to examine how these factors are affected by the projection of objects within the AR system.

2. **Drone Appearance Evaluation:** Focused on drone appearance, referencing previous research [23]. These items assess the extent of changes in the drone's appearance due to AR-based appearance modification.

3. **Subjective Evaluation of Operability:** Focused on the subjective evaluation of operability, referencing previous research [15]. These elements investigate how AR-based appearance modification affect the operational feel of the drone.

## Results

This section presents the analysis of drone operation tasks and the questionnaire conducted described in Section Proposed Methodology to clarify the effects of AR-projected objects on drone impressions and operational behavior.

### Analysis of drone operation tasks

This subsection explains the evaluation methods for drone operation tasks and their analysis. Using the method described in Section Creation of Projected Objects, the direction and magnitude of the controller operations were detected to extract the operating characteristics. Based on the controller operation data collected, the operational metrics were derived and summarized in Table 1. These metrics were used to quantitatively evaluate the impact of AR-projected objects on operational behavior.

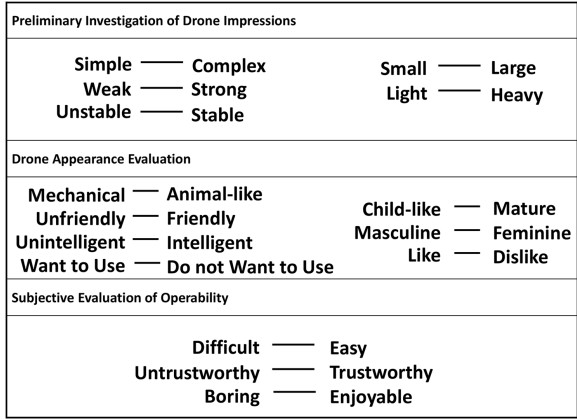

**Fig 11. Questionnaire items.** This figure illustrates the questionnaire items used in this study, categorized into three groups. Participants respond to these items using a 7-point Likert scale.

**Table 1. List of metrics and definitions.**

| Metric | Definition |
| --- | --- |
| Total Number of Operations | Number of complete sets of increase and decrease in controller movement magnitude $L$ |
| Operations per Unit Time | Number of operations per unit time |
| Average Maximum Control Displacement | Average of maximum displacement per operation |
| Mean Operation Size 1 | Average value of $L$ classified into [0.1, 0.3, 0.5, 0.7, 1.0] |
| Mean Operation Size2 | Average value of $L$ classified into [0.1, 0.2, 0.4, 0.7, 1.0] |
| Average Duration per Operation | Average duration of each operation |
| Average Size per Unit Duration | Mean value of operation size divided by operation duration |
| Number of Direction Reversals | Number of times the operation direction was reversed |
| Idle Time | Total duration of idle periods |
| Mean Controller Operation Speed | Average speed of controller movement |
| Mean Controller Operation Acceleration | Average acceleration of controller movement |
| Cumulative Displacement | Cumulative displacement of operation size |
| Movement Smoothness | Smoothness evaluated using variance in frame-by-frame velocity changes |
| Number of Continuous Movements | Number of continuous movements performed within a short time (0.3 seconds) |
| Average Time Interval Between Actions | Average interval between operations |
| Variation in Maximum Control Displacement | Standard deviation of maximum displacement per operation |
| Number of Repeated Movements | Number of repeated operations |
| Total Time | Total duration of task execution |
| Number of Collisions (Pole) | Number of times the drone collided with poles |
| Number of Collisions (Wall) | Number of times the drone collided with walls |
| Crash | Number of drone crashes |
| Number of Screen Exits | Number of times the drone exited the screen |

Factor analysis revealed three meaningful dimensions of operational behavior, which together explained a large proportion of the variance (99.1%). The factor loading matrix is provided in Supporting Table A in S1 File. Each factor was interpreted and named based on the dominant contributing metrics.

**Operational Frequency and Movement Volume (Factor 1):** Metrics related to frequency and movement volume (e.g., Total Number of Operations, Operations per Unit Time, Average Time Interval Between Actions).

**Control Size and Instability (Factor 2):** Metrics related to control size and movement instability (e.g., Mean Operation Size, Movement Smoothness, Number of Direction Reversals).

**Task Performance (Factor 3):** Metrics related to task performance (e.g., total time, crash, number of exits from screen).

## Questionnaire analysis

Factor analysis revealed three interpretable dimensions of drone impressions. Together, these factors explained a substantial portion of the variance (64.1%). Detailed factor loadings are reported in Supporting Table B in S1 File. Factors were labeled according to their primary item loadings.

**Emotional evaluation (Factor 1):** Negative emotional evaluations and acceptance (e.g., likeability, fun, friendliness).

**Appearance and Social Evaluation (Factor 2):** Impressions of drone appearance and social evaluation (e.g., Age, Animal Likeness, Gender).

**Physical characteristics (Factor 3):** Physical characteristics of the drone (e.g., Size, Weight).

A comparison of factor scores between Conditions A and B was performed using paired $t$ tests or Wilcoxon signed rank tests depending on normality, as verified by the Shapiro-Wilk test. The results are summarized in Table 2. Significant differences were observed for Factor 2 (Appearance and Social Evaluation) and Factor 3 (Physical Features), but not for other factors.

To examine the magnitude of differences, Hedges' $g$ and 95% confidence intervals were computed for all questionnaire items [30]. The full results are provided in Supporting Table C in S1 File.

Effect size analysis reinforced this pattern: operational measures showed negligible effects, whereas several impression-related measures showed meaningful effects.

## Analysis of open-ended questionnaire responses

In addition to the questionnaire items shown in Fig 11, the participants were asked to respond to the following open-ended questions. These responses were collected from 10 participants under both experimental conditions A and B:

• "Please freely describe your impressions of the drone's appearance after operating it."

• "Please freely describe your impressions of the drone's usability after operating it."

The responses were divided into individual sentences and each sentence was labeled to clarify "what it refers to." The labels were assigned based on the following categories:

• **Appearance**: Opinions on the appearance of the drone.

• **Safety**: Impressions or concerns about safety.

• **Usability**: Opinions on operability or ease of control.

• **Visibility**: Opinions on visibility or depth perception.

• **Sound**: Opinions on the drone's sound.

Each category was also assigned a sentiment label: **Negative** (critical opinions), **Positive** (favorable opinions), or **Neutral** (neutral opinions). The following are examples of the labeling process:

• "Mechanical"→**Appearance-Negative**

• "The bird illustration was cute"→**Appearance-Positive**

• "It moved more than I expected, making fine adjustments difficult"→**Usability-Negative**

The distribution of labels under Conditions A and B is shown in Fig 12. Fisher's exact test was conducted to compare label frequencies between conditions (Table 3). Differences between conditions emerged primarily in appearance-related

**Table 2. Comparison of factor scores between conditions A and B.**

| Factor | Mean A | Mean B | p-value |
|---|---|---|---|
| Factor1_Operation | 0.121 | −0.121 | 0.333 |
| Factor2_Operation | 0.020 | −0.020 | 0.803 |
| Factor3_Operation | 0.102 | −0.102 | 0.272 |
| Factor1_Survey | 0.014 | −0.014 | 0.873 |
| Factor2_Survey | −0.620 | 0.620 | $3.81 \times 10^{-5}$ |
| Factor3_Survey | 0.325 | −0.325 | 0.00025 |

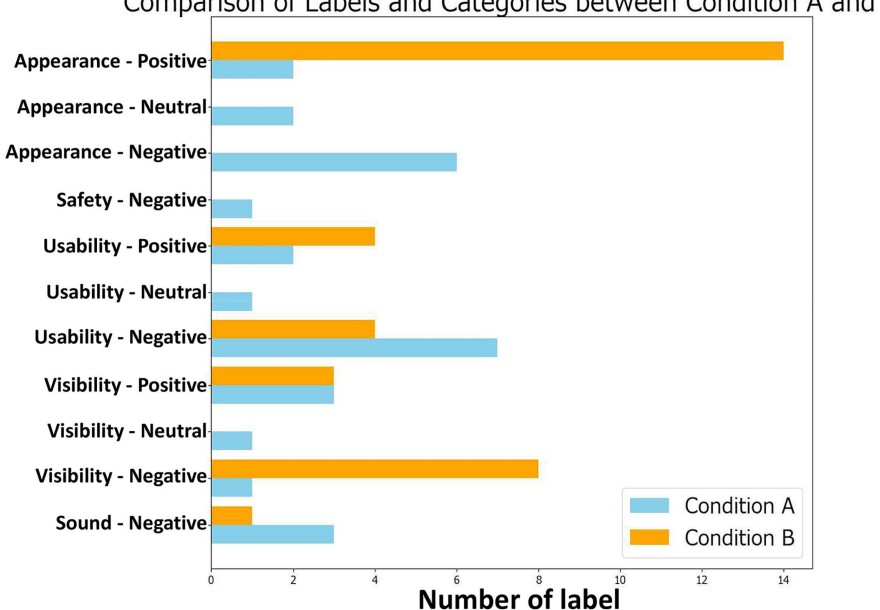

**Fig 12. Distribution of sentiment labels in open-ended responses under Conditions A and B.**

**Table 3. Comparison of labels in open-ended questionnaire responses for conditions A and B.**

| Category | Odds Ratio | p-value |
|---|---|---|
| Appearance | 0.000 | 0.000375 |
| Safety | N/A | 1.000000 |
| Usability | 0.286 | 0.334842 |
| Visibility | 8.000 | 0.235165 |
| Sound | N/A | 1.000000 |

impressions, while other aspects remained largely similar. For Safety and Sound, odds ratios could not be calculated due to zero counts in one or both conditions.

## Discussion

One possible reason for the lack of impact on controller operation is that the projected object was a 2D image. This may have reduced depth perception and counteracted potential positive effects on controller usability. Specifically, while the 2D image visually emphasized the drone's appearance, it may have decreased visual cues for understanding the drone's spatial positioning, thus limiting its effect on operational behavior.

These findings suggest that AR-based appearance modification can enhance social evaluation and perceived physical characteristics of drones without degrading operational performance. In other words, visual augmentation can improve how drones are perceived while maintaining stable controllability.

This interpretation aligns with prior industrial AR/VR research, where visual augmentation has been shown to enhance user perception and interaction without necessarily altering core task performance [7,8]. Our results extend this understanding to fast-moving robotic agents, demonstrating that similar principles apply even in dynamic human–drone interaction contexts.

This study was conducted in a controlled environment using a green screen and color markers to ensure stable detection and safety. Although such conditions may limit ecological validity, they enabled systematic evaluation of AR-based appearance modification on moving drones.

Similar to Umar (2022) [31], who validated a safety assessment tool under controlled conditions before proposing real-world application, this research represents an initial step toward practical deployment.

As AR device capabilities and real-time processing technologies continue to improve, the proposed method could be extended to outdoor or real-world drone operation environments. Future work should also investigate 3D or volumetric projections, which may provide richer spatial cues and potentially influence operational behavior as well as impressions.

One limitation of this study is the use of 2D AR-based appearance modification rather than fully spatial 3D AR representations. Because 2D projections provide limited depth cues, they may have reduced the perceptual realism of the projected objects and partially contributed to the limited effects on operational behavior. Although spatial 3D AR was initially considered, real-time rendering on fast-moving drones was not feasible with the computational performance of available wearable devices at the time of development. Future research using spatially registered 3D AR could clarify whether richer depth information leads to stronger influences on both perception and operation.

## Conclusion

The appearance of conventional drones has been found to provoke fear and make it difficult to discern their direction of motion. However, due to the challenge of dynamically altering the appearance of fast-moving drones in real time, little attention has been paid to how changes in drone appearance might affect user impressions or operational behavior.

In response, this study posed the research question: **"Can changing the appearance of drones through AR influence user operational behavior and impressions of drones?"**

We proposed a method to project objects onto drones using AR, altering their appearance, and conducted experiments to evaluate their impact.

The results did not show significant differences in factors related to operational behavior between the two conditions. However, significant differences were observed in factors related to social evaluation, physical characteristics, and appearance based on both the questionnaire results and free text responses.

These findings suggest that AR-projected objects can improve impressions of drones' social evaluation, physical features, and appearance without adversely affecting controller operation. The proposed method can potentially reduce psychological burdens on users while maintaining their operational performance.

This approach could help bridge the psychological gap between drones and the public, enhancing the social acceptance of drone technology. It offers a promising avenue for integrating drones into urban environments, fostering a greater sense of familiarity and trust among residents, and expanding the social adoption of this technology.

As a near-term extension, future work will focus on further validation of the present findings through experiments with more rigorous and controlled designs. In addition, the potential of spatially registered 3D AR projections in real-world environments will be explored. As highlighted by Al-Salem et al. (2025) [32], technical features such as camera mobility and spatial perception are critical in drone-based safety applications. While the present study employed 2D projections, 3D visualization using devices such as HoloLens may enhance spatial awareness, whereas camera-based systems such as Oculus may provide more limited 3D benefits.

In parallel, near-term studies will examine how specific visual attributes (e.g., color, shape, and texture) and different AR-projected objects influence user perception, social evaluation, and psychological responses such as anxiety or fear.

As longer-term research directions, future work will investigate user behavior under prolonged exposure and in more complex operational contexts, including multi-drone operations and shared-space scenarios. From a technical perspective, further research will explore integration with drone sensors for dynamic AR adaptation, predictive object placement

 

algorithms, 3D surface projections, and multi-user support. These efforts aim to enhance the practical applicability and generalizability of AR-modified drone systems in real-world deployments.

## Supporting information

**S1 File. Supporting tables.** This file contains additional tables supporting the results of this study. • Table A. Factor loadings for operational metrics. • Table B. Factor loadings for questionnaire items. • Table C. Effect sizes (Hedges' g) and 95% confidence intervals.
(PDF)

## Acknowledgments

This study is partially supported by JST Moonshot R&D Program (AI-ENGAGE), Grant Number JPMJMS25E4, Japan.

## Author contributions

**Conceptualization:** Runa Sawayama, Hajime Nobuhara.

**Data curation:** Runa Sawayama.

**Formal analysis:** Runa Sawayama.

**Funding acquisition:** Hajime Nobuhara.

**Investigation:** Runa Sawayama.

**Methodology:** Runa Sawayama.

**Project administration:** Hajime Nobuhara.

**Software:** Runa Sawayama.

**Supervision:** Hajime Nobuhara.

**Validation:** Runa Sawayama.

**Visualization:** Runa Sawayama.

**Writing – original draft:** Runa Sawayama.

**Writing – review & editing:** Hajime Nobuhara.

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
