## [Decision Letter · Decision Letter 0]

29 Sep 2025

Dear Dr. Sawayama,

Thank you for submitting your manuscript to PLOS ONE. After careful consideration, we feel that it has merit but does not fully meet PLOS ONE’s publication criteria as it currently stands. Therefore, we invite you to submit a revised version of the manuscript that addresses the points raised during the review process.

Apart from addressing the reviewers comments, make sure you address the following points.

1. The study includes only 21 participants, limiting generalizability. Additional details on participant demographics (age, gender balance, prior drone/AR experience) would strengthen the context for interpreting results. In order to justify the participants size, I suggest to review this article "Perceptions on safety climate: a case study in the Omani construction industry".

2. The discussion acknowledges the use of 2D projections, but more analysis of how 3D projections or real-world deployment might alter findings would add depth. See this recent study which consider a number of technical features "Applications of drones for safety inspection in the Gulf Cooperation Council construction".

3. The highly controlled laboratory environment (green screen, stickers) may not reflect real-world drone operation. A clearer discussion of ecological validity and potential field applications would enhance impact. Suggest to review this article for validation "Developing a safety climate assessment tool for Omani construction industry"

4. While factor analysis and t/Wilcoxon tests are described, reporting effect sizes (e.g., Cohen’s d) and confidence intervals would improve transparency and allow readers to assess practical significance. I suggest to review this document which provide a number statistical analysis which could be useful in your study "Developing toolkits and guidelines to improve safety performance in the construction industry in Oman"

We look forward to receiving your revised manuscript.

Kind regards,

Tariq Umar, PhD, CEng, EUR ING, MICE, FHEA

Academic Editor

PLOS ONE

Journal Requirements:

3. Please ensure that you refer to Figures 1 and 3 in your text as, if accepted, production will need this reference to link the reader to the figures.

Additional Editor Comments:

Dear Authors,

Your paper has been reviewed by two independent reviewers and have recommended a major revision. I therefore invite you to address the reviewers comments along with the editor comments and submit a revise version of the paper for consideration. Apart from addressing the reviewers comments, make sure you address the following points.

1. The study includes only 21 participants, limiting generalizability. Additional details on participant demographics (age, gender balance, prior drone/AR experience) would strengthen the context for interpreting results. In order to justify the participants size, I suggest to review this article "Perceptions on safety climate: a case study in the Omani construction industry".

2. The discussion acknowledges the use of 2D projections, but more analysis of how 3D projections or real-world deployment might alter findings would add depth. See this recent study which consider a number of technical features "Applications of drones for safety inspection in the Gulf Cooperation Council construction".

3. The highly controlled laboratory environment (green screen, stickers) may not reflect real-world drone operation. A clearer discussion of ecological validity and potential field applications would enhance impact. Suggest to review this article for validation "Developing a safety climate assessment tool for Omani construction industry"

4. While factor analysis and t/Wilcoxon tests are described, reporting effect sizes (e.g., Cohen’s d) and confidence intervals would improve transparency and allow readers to assess practical significance. I suggest to review this document which provide a number statistical analysis which could be useful in your study "Developing toolkits and guidelines to improve safety performance in the construction industry in Oman"

Reviewers' comments:

Reviewer's Responses to Questions

**Comments to the Author**

1. Is the manuscript technically sound, and do the data support the conclusions?

Reviewer #1: Partly

Reviewer #2: Partly

2. Has the statistical analysis been performed appropriately and rigorously?

Reviewer #1: N/A

Reviewer #2: N/A

3. Have the authors made all data underlying the findings in their manuscript fully available?

Reviewer #1: Yes

Reviewer #2: Yes

4. Is the manuscript presented in an intelligible fashion and written in standard English?

Reviewer #1: No

Reviewer #2: Yes

Reviewer #1: 1. This article investigates the use of AR for object appearance assessment. The work however, has rooms to improve before it can be considered for possible publication with the journal PLOS One.

2. A large number of participants are engaged in the experiment designed. Better control and experiment design should be conducted to provide more objective comparison.

3. The novelty of the study should be much better highlighted in the abstract. Contributions and limitations of the research should be better discussed as well in the Conclusion.

4. There are 25 reference papers cited in the article which is in the low end. Among 25 reference papers, only 6 of them are published after 2020.

5. The accuracy of the AR projection or alignment is something of interesting that should be discussed in more detail. Also, drone flighting is a dynamic process highly related to robotic path planning. Some further elaboration is expected.

6. Most of the photos should be rotated by 90 degree clockwise to allow easier reading.

7. The state-of-the-art below can help readers better understand the complexity of the research:

- Xu P., et al (2025). SafeSpect: Safety-First Augmented Reality Heads-up Display for Drone Inspections. CHI 2025, Yokohama, Japan.

- Souravik D. et al (2020), Automatic re-planning of lifting paths for robotized tower cranes in dynamic BIM environments, Automation in Construction 110, 102998, ISSN 0926-5805, https://doi.org/10.1016/j.autcon.2019.102998.

Reviewer #2: While the study is intriguing, still there are several aspects that need to be strengthened for the paper to be suitable for publication. Below are detailed suggestions that may improve the manuscript:

The authors should clarify the novelty of the work. Since, the studies are already existing for the selected problems. So, the authors should state clearly about the unique work. Also, the authors should clarify the challenges with the existing practices for the chosen problem.

Comparative analysis is needed to explore the uniqueness of the proposed work. Since, the existing literature is exploring similar studies.

The motivation of the work is not clear in the introduction section. I suggest the authors restructure and emphasize the proposed work.

Please clarify, how does AR-modified drone appearance influence user perceptions of drone friendliness and approachability? Which visual attributes (color, size, shape, texture) most significantly impact user operational behavior?

How do different AR-projected objects affect the perceived personality or intent of drones? Does AR-modified appearance affect users’ trust in drones during navigation and task execution?

How do demographic factors (age, gender, cultural background) influence perception and behavior toward AR-modified drones? Can AR modifications reduce user anxiety or fear associated with drones in close proximity?

How does AR appearance impact users’ willingness to interact with drones in shared spaces? Are there measurable differences in drone operation efficiency (e.g., speed, accuracy) when AR-modified?

How does repeated exposure to AR-modified drones influence long-term user behavior and perception? Do AR modifications affect social evaluation factors such as anthropomorphism or animal likeness?

Can AR appearance adjustments increase drone acceptance in sensitive environments (e.g., schools, hospitals)? How do users perceive risk differently with different AR-modified drone appearances?

Does AR modification affect user cooperation in multi-drone environments? Are there differences in task performance or errors when drones carry different AR projections?

Can AR appearance serve as a communication tool for drone intentions (e.g., signaling stopping, direction changes)? Which AR projection methods (headset-based, screen-based, projector-based) provide the most stable drone visualization?

How can AR projections maintain alignment with drones in high-speed motion? What are the latency requirements for real-time AR appearance updates on moving drones?

How can drone sensors and AR systems be integrated for dynamic appearance adaptation? What algorithms can predict optimal AR object placement for improved user perception?

How does environmental lighting affect AR object visibility and realism on drones? What metrics can quantify the accuracy of AR projection on 3D drone surfaces?

How can AR projection systems handle multiple users with different viewpoints simultaneously? What hardware and computational requirements are needed for real-time rendering of AR objects?

Stated Literature is shallow. The authors are suggested to review more new and relevant research to support their research contribution. The recent state of art keywords related to other immersive technologies; Challenges and opportunities on AR/VR technologies for manufacturing systems in the context of industry 4.0, Augmented reality-based guidance in product assembly and maintenance/repair perspective: A state of the art review on challenges and opportunities. Augmented reality guided autonomous assembly system: A novel framework for assembly sequence input validations and creation of virtual content for AR instructions development.

The relevant other keywords: Augmented reality aided object mapping for worker assistance/training in an industrial assembly context: AR/VR assisted integrated framework of autonomous disassembly system for industrial products

How can the system detect and correct misalignment or drift of AR objects on moving drones? Can machine learning optimize AR appearance based on user feedback or interaction data?

How can AR modifications be standardized across different drone sizes and models? How can AR systems balance visual fidelity with battery consumption and drone flight performance?

What methods can assess the real-time influence of AR projections on user behavior? How can AR systems dynamically adjust drone appearance to avoid occlusions or visual interference in complex environments?

.

Reviewer #1: No

Reviewer #2: **Yes:** Eswaran MEswaran MEswaran MEswaran M

---

## [Author Response · Author response to Decision Letter 1]

25 Nov 2025

A detailed response to all reviewer and editor comments has been uploaded as a separate file.

Please refer to the attached file titled “Response to Reviewers.”

---

## [Decision Letter · Decision Letter 1]

29 Dec 2025

Dear Dr. Sawayama,

Thank you for submitting your manuscript to PLOS ONE. After careful consideration, we feel that it has merit but does not fully meet PLOS ONE’s publication criteria as it currently stands. Therefore, we invite you to submit a revised version of the manuscript that addresses the points raised during the review process.

We look forward to receiving your revised manuscript.

Kind regards,

Tariq Umar, PhD, CEng, EUR ING, MICE, FHEA

Academic Editor

PLOS One

Journal Requirements:

Additional Editor Comments :

Dear Author(s),

The revised paper has now been reviewed by two independent reviewers. You can see one reviewer has recommended the paper but the other one have some additional comments. I therefore invite you to consider these comments and revise the paper accordingly.

Reviewers' comments:

Reviewer's Responses to Questions

**Comments to the Author**

Reviewer #2: All comments have been addressed

Reviewer #3: (No Response)

2. Is the manuscript technically sound, and do the data support the conclusions?

Reviewer #2: Yes

Reviewer #3: Partly

3. Has the statistical analysis been performed appropriately and rigorously?

Reviewer #2: Yes

Reviewer #3: N/A

4. Have the authors made all data underlying the findings in their manuscript fully available?

Reviewer #2: Yes

Reviewer #3: No

5. Is the manuscript presented in an intelligible fashion and written in standard English?

Reviewer #2: Yes

Reviewer #3: Yes

Reviewer #2: Comments are addressed by the authors. No more comments. manuscript improved in the current version. This can be accepted.

Reviewer #3: 1. The manuscript addresses an interesting and underexplored topic: the influence of AR-based appearance modification on drone perception and operation.

2. The experimental setup is carefully designed, ethically approved, and well documented.

3. The study provides useful empirical evidence showing that AR-projected objects influence social and appearance-related perceptions without negatively affecting operational performance.

4. The use of AR to dynamically modify the appearance of real, fast-moving drones represents a novel contribution compared to prior AR–drone studies that focus mainly on interfaces or virtual drones.

5. The combination of quantitative controller-operation metrics with subjective and open-ended perceptual evaluations strengthens the contribution.

6. The manuscript would benefit from a clearer articulation of its novelty relative to prior AR-based drone visualization and acceptance studies in the Introduction and Discussion. Please refer:: Challenges and opportunities on AR/VR technologies for manufacturing systems:: AR/VR assisted integrated framework of autonomous disassembly system.

7. The experimental design, task structure, counterbalancing, and choice of metrics are appropriate for the stated research question.

8. The use of factor analysis is justified; however, the manuscript reports extensive factor loadings and statistical details that may not all be necessary in the main text.

9. The limited sample size (n = 21) is acceptable for an exploratory study but should be more clearly framed as such earlier in the manuscript.

10. The Results section is comprehensive but overly detailed, with repeated descriptions of statistical outcomes across text, tables, and figures.

11. Several tables (e.g., detailed factor loadings and effect-size tables) could be moved to Supplementary Material to improve readability.

12. Greater emphasis on high-level trends rather than exhaustive numerical reporting would strengthen clarity.

13. The Discussion provides reasonable explanations for why AR-based appearance modification affected perception but not operational behavior.

14. The potential influence of using 2D projections instead of 3D AR could be discussed more explicitly as a limiting factor.

15. The comparison with related work is appropriate, but the implications for real-world drone deployment could be synthesized more concisely.

16. The Conclusion appropriately summarizes the findings but could be shortened by reducing methodological repetition.

17. Future work is well motivated; however, it may be beneficial to more clearly separate near-term extensions (e.g., 3D AR projection) from long-term research directions (e.g., multi-drone scenarios).

18. The manuscript is generally well written, though minor grammatical errors and formatting inconsistencies are present.

19. Repetition of certain sentences and phrases should be reduced, particularly in the Introduction and Results sections.

20. Terminology should be reviewed for consistency (e.g., “appearance modification,” “object projection,” “AR projection”).

.

Reviewer #2: No

Reviewer #3: No

---

## [Author Response · Author response to Decision Letter 2]

12 Feb 2026

We have provided detailed responses to all reviewer and editor comments in the “Response to Reviewers” file. Please refer to that document for our responses.

---

## [Decision Letter · Decision Letter 2]

25 Mar 2026

Enhancing user perception and drone flight control through AR-based object projection

PONE-D-25-33316R2

Dear Dr. Sawayama,

We’re pleased to inform you that your manuscript has been judged scientifically suitable for publication and will be formally accepted for publication once it meets all outstanding technical requirements.

Kind regards,

Ajay Kumar Vyas

Academic Editor

PLOS One
---

## [Editor Report · Acceptance letter]

PONE-D-25-33316R2

PLOS One

Dear Dr. Sawayama,

I'm pleased to inform you that your manuscript has been deemed suitable for publication in PLOS One. Congratulations! Your manuscript is now being handed over to our production team.

Kind regards,

on behalf of

Dr. Ajay Kumar Vyas

Academic Editor

PLOS One